# Co-benefits of protecting mangroves for biodiversity conservation and carbon storage

Md Mizanur Rahman [1,2,3], Martin Zimmer[4,5], Imran Ahmed[6], Daniel Donato[7], Mamoru Kanzaki[3] & Ming Xu[1,2 ✉]

The conservation of ecosystems and their biodiversity has numerous co-benefits, both for local societies and for humankind worldwide. While the co-benefit of climate change mitigation through so called blue carbon storage in coastal ecosystems has raised increasing interest in mangroves, the relevance of multifaceted biodiversity as a driver of carbon storage remains unclear. Sediment salinity, taxonomic diversity, functional diversity and functional distinctiveness together explain 69%, 69%, 27% and 61% of the variation in above- and belowground plant biomass carbon, sediment organic carbon and total ecosystem carbon storage, respectively, in the Sundarbans Reserved Forest. Functional distinctiveness had the strongest explanatory power for carbon storage, indicating that blue carbon in mangroves is driven by the functional composition of diverse tree assemblages. Protecting and restoring mangrove biodiversity with site-specific dominant species and other species of contrasting functional traits would have the co-benefit of maximizing their capacity for climate change mitigation through increased carbon storage.

[1] Henan Key Laboratory of Earth System Observation and Modeling, Henan University, Kaifeng, China. [2] College of Geography and Environmental Science, Henan University, Kaifeng, China. [3] Graduate School of Agriculture, Kyoto University, Kitashirakawa Oiwake, Sakyo-ku, Kyoto, Japan. [4] Leibniz Centre for Tropical Marine Research & University of Bremen, Bremen, Germany. [5] IUCN-SSC Mangrove Specialist Group, Gland, Switzerland. [6] Bangladesh Forest Department, Bon Bhaban, Dhaka, Bangladesh. [7] School of Environmental and Forest Sciences, University of Washington, Seattle, WA, USA. ✉email: mingxu@henu.edu.cn

The role of forests in carbon sequestration renders their protection a cost-effective nature-based solution for reducing atmospheric concentration of carbon dioxide[1,2]. How this carbon sequestration property is influenced by human-induced changes in forest biodiversity has led to hundreds of studies, mainly in terrestrial ecosystems[3–7] (temperate, boreal, tropical, subtropical, agroforestry and plantation forests) and a few mangrove studies[8–13]. Biodiversity-carbon research[3,4] in terrestrial ecosystems has spanned trait-based (morphological, phenological, physiological and chemical characteristics of plants) biodiversity (e.g., functional composition, community-level mean value of plant traits and functional diversity, variation of species traits within a community) along with conventional diversity (species and taxonomic diversity) approaches. However, biodiversity-carbon studies in mangroves, among the most carbon-rich ecosystems in the tropics[14–16], are still essentially restricted to conventional diversity[8–10,14,17,18]. This reveals a crucial knowledge gap for understanding mangroves' contribution to mitigating climate change, as functional diversity studies can elucidate the mechanisms behind species-diversity association with carbon storage[4,19]. Filling this knowledge gap will inform mangrove conservation policy, such as REDD + (Reducing Emissions from Deforestation and forest Degradation plus), the Post-2020 biodiversity agenda of the Convention of Biological Diversity, and mangrove rehabilitation and afforestation in general[20].

Previous studies have shown that species diversity has both a direct and indirect association with ecosystem processes through both functional diversity and functional composition, (i.e., the functional distinctiveness of taxonomic units and of the community), on carbon storage in terrestrial ecosystems[4,19,21]. The association of these components of biodiversity with productivity or carbon storage have usually been explained by two hypotheses: niche complementarity and selection[4]. The niche complementarity hypothesis suggests that the capacity to store carbon in a community is primarily determined by coexisting species with greater trait variance[4,21]. On the other hand, the selection theory assumes that the most abundant species with their key functional traits determine the ecosystem-wide carbon storage[4,21]. Thus, if species diversity or functional diversity influences carbon storage, this would lend support to the niche complementarity hypothesis as a mechanism for the storage of blue carbon. Conversely, if functional composition influences the storage of blue carbon, this lends support to the selection hypothesis[4]. Alternatively, both mechanisms may jointly control carbon storage in some tropical forests, including blue carbon storage if both species richness or functional diversity and functional composition influence mangrove carbon storage[4,22].

Plant traits related to carbon storage are used to elucidate the relationship of functional diversity and functional composition with carbon storage[23]. For instance, wood density, maximum canopy height, specific leaf area, leaf dry matter content, leaf photosynthesis rate, leaf carbon and nitrogen content and their ratios are commonly used plant traits in functional ecological studies[3,4,24]. Different functional traits occupy different niche axes and determine the growth, productivity, stability and resistance of overall carbon storage in a community[3,25]. For example, high wood density provides mechanical support, affects water and nutrient supply and hence plays a key role in plant growth, survival and forest carbon stock[26]. Maximum canopy height, an indicator of light interception[27] that reflects the relative resource-use capacity of a species in a community, determines community scale biomass allocation and carbon storage[3,4,28–30]. Plant traits such as leaf carbon, nitrogen and their ratio, leaf dry matter content or contents of phenolic compounds, as indicators of litter quality for decay and decomposition, could influence nutrient

cycling and productivity[31–34]. For instance, higher litter nitrogen content or phenolic content could enhance microbial litter decay and drive sediment carbon content and productivity in mangroves[34–36].

The carbon fixed by mangrove plants, and accumulated in biomass and sediment organic matter over time, is influenced by integrated mechanisms involving both abiotic and biotic influences[37,38]. On a wider geographical scale, climatic factors (abiotic) including temperature and precipitation have an association with carbon storage[9,30], while the local or regional hydrological system, such as freshwater flow or the tidal regime, controls porewater salinity in the sediment[9,37,39,40], which, in turn, influences plant growth, stand structure and ecosystem productivity[41] and therefore blue carbon storage. Sediment salinity also controls total sediment nutrients[39] (nitrogen, phosphorus and total organic matter), as well as the distribution of the plant species[42] and, hence, their functional traits. For example, low-salinity areas are rich in plant species with a higher potential for photosynthetic nitrogen use efficiency[43], resulting in higher dry matter content and higher growth form (maximum canopy height) compared to plant species in high-salinity areas. Salinity-driven variation of species richness and composition can also affect two other components of biodiversity that are based on plant functional traits: functional diversity and functional composition of different plant traits in a community. As a result, sediment salinity may also have indirect associations with blue carbon storage in mangroves through these three components of biodiversity: species richness, functional diversity and functional composition. The integration of the functional composition of the community, (i.e., the functional distinctiveness of species within the community), into modelling along with other traditional and modern diversity measures and sediment salinity can provide new insights into the relationship between mangrove biodiversity and blue carbon storage.

Here we explore the relationships among species richness, taxonomic diversity, functional diversity and functional composition with aboveground and belowground plant biomass carbon, sediment organic carbon and total ecosystem carbon storage after considering for the association of sediment salinity, using a structural equation model (a robust approach compared to multiple linear or bivariate relationship, as it considers direct, indirect and total association of predictors[44]) in the mangroves of the Sundarbans Reserved Forest, Bangladesh. In this study, we analyzed a large-scale forest carbon inventory dataset which includes 90 forest plots distributed across the forest area (Supplementary Table 1 provides a description of field data including sediment salinity). Each plot is composed of five 10 m radius circular subplots[45]. We estimated different biomass carbon pools, using both locally developed species-specific and common allometric equations for mangroves[45–48]. For estimating sediment organic carbon, we used the organic carbon storage of the top 1 m (Supplementary Table 1; methods[45]). We used maximum canopy height, wood density, photosynthesis rate, and four-leaf litter qualities[27] (carbon, nitrogen, C:N ratio, and dry matter content; Supplementary Table 2) as plant functional traits. For species diversity, we used species richness (count of species per plot) and Shannon diversity. We calculated the functional dispersion and community weighted mean as proxies for functional diversity and functional composition, respectively. We focus on two questions: (i) how are different carbon pools in mangroves related to species diversity, functional diversity and functional composition, after accounting for direct and indirect associations of the key gradient of sediment salinity? and (ii) how can these relationships guide conservation and rehabilitation or afforestation policies for mangroves? For answering the first question, we hypothesized that (i) sediment salinity has direct and indirect associations with

blue carbon storage through species diversity, functional diversity and functional composition; (ii) species diversity has direct and indirect associations with blue carbon storage through functional diversity and functional composition; (iii) functional composition has a greater association with blue carbon storage than the other two biodiversity components. Species richness and functional distinctiveness of wood density, maximum canopy height and leaf litter nitrogen are positively associated with blue carbon storage where functional distinctiveness, had the strongest association with blue carbon storage. Our findings indicate that blue carbon storage in mangroves can be best sustained through compositionally and functionally diverse tree assemblages.

## Results

**Bivariate relationships among different covariates**. Bivariate analyses (to observe independent associations of all predictors with different blue carbon storage, apart from our structural equation model) revealed that aboveground and belowground plant biomass carbon, sediment organic carbon and total ecosystem carbon storage decreased with increasing sediment salinity (Fig. 1a–d, $P < 0.05$). Species richness (measured by counting observed species in each plot) and functional composition of wood density and maximum canopy height had positive associations with aboveground and belowground plant biomass carbon, sediment organic carbon (although maximum canopy height had no significant effect; $P > 0.05$) and total ecosystem carbon storage (Fig. 1i–p). However, the functional composition component of leaf litter nitrogen content and functional diversity component of leaf litter dry matter content did not have any significant association with any of the blue carbon storage (Fig. 1q–x; $P > 0.05$). Separately, sediment salinity, species richness, functional composition of wood density and maximum canopy height explained less variance in different blue carbon storage (Fig. 1a–p). All components of biodiversity (except species richness, $P > 0.05$) decreased significantly with increasing sediment salinity (Fig. 2a–e). Species richness had no significant association with functional diversity or the functional composition factor of wood density (Fig. 2f and i), but we observed a significant negative and positive association, respectively, with functional composition aspects of maximum canopy height and the nitrogen content of the leaf litter (Fig. 2e; $P < 0.05$). With increasing functional distinctiveness (depicted by functional composition) of maximum canopy height, the functional distinctiveness of wood density increased, while the functional distinctiveness of leaf litter characteristics showed a decreasing trend (Fig. 2j and k).

**Drivers of blue carbon storage in mangrove**. While bivariate analyses only show one-way associations of sediment salinity and multiple biodiversity measures with different blue carbon storage, structural equation models show interactive associations of these factors with different blue carbon storage (Fig. 3). Sediment salinity had no significant direct association with blue carbon storage, but had an indirect association with them through the functional composition of different traits (Fig. 3). Four structural equation models for aboveground and belowground plant biomass carbon, sediment carbon and total ecosystem carbon storage fit the datasets well, indicating no significant deviation between observed datasets and models (Fig. 3), as indicated by a Chi-square test statistic of $\chi^2 = 6.83$, $P = 0.093$, a comparative fit index close to 1 (CFI = 0.99), and a standardized root mean square residual close to 0 (SRMR = 0.03; Supplementary Table 5). Sediment salinity, species richness, functional composition of wood density, maximum canopy height and leaf litter nitrogen, as well functional diversity of leaf litter dry matter content together

explained 69%, 69%, 26 % and 61% of variation of aboveground and belowground plant biomass carbon, sediment carbon and total ecosystem carbon storage, respectively (Fig. 3). The functional diversity and functional composition of the other traits were strongly correlated and their variance inflation factors were >10 and thus they were not retained in the structural equation models (Supplementary Figs. 1–2; Supplementary Tables 3–4; methods).

For aboveground and belowground biomass carbon, the strong negative association of sediment salinity was mediated by the functional composition of wood density, while for sediment organic carbon and total ecosystem carbon it was mediated strongly by the functional composition of maximum canopy height (Table 1). On the other hand, the greatest association of species richness was induced by increasing the functional composition of litter nitrogen on aboveground and belowground biomass carbon (Table 1). However, for sediment organic carbon and total ecosystem carbon, the direct associations of species richness were the strongest paths in the structural models (Fig. 3c and Table 1). When we considered all direct associations, including sediment salinity, species richness, functional composition and functional diversity, functional composition always had the strongest associations with different types of blue carbon storage, with the association size depending on the type of traits, while functional diversity had the lowest association (Fig. 3). For instance, the functional composition of wood density was the strongest driver in the case of aboveground plant biomass carbon, whereas the functional composition of both wood density and maximum canopy height were the strongest drivers for belowground plant biomass carbon (Fig. 2a, b).

## Discussion

We quantified the interacting associations of species richness, functional diversity and functional composition of mangrove forests with different blue carbon storage in the Sundarbans Reserved Forest, one of the largest mangrove forests in the world and among the largest carbon deposits in the tropics[16]. We found that species richness and functional composition had a positive relationship with blue carbon storage, while functional diversity had a negative association after considering the association of sediment salinity in structural equation models. Sediment salinity mainly decreased functional diversity and distinctiveness of species but not species richness, and hence, its associations with different blue carbon storage are mediated by functional diversity and functional composition of the community. Species richness had positive indirect associations with plant above- and below-ground biomass carbon through functional composition, while it had both direct and indirect (through functional composition) associations with sediment organic carbon and total ecosystem carbon. Our results also indicate that among the multiple facets of biodiversity, functional composition exerted the strongest and functional diversity the weakest -associations with different blue carbon storage. From this study, we derive several key implications for community ecology in mangroves and the conservation value of mangroves for climate change mitigation.

First, the positive association of species richness on different blue carbon storage contributes to answering the unexplored question of whether increasing species richness in relatively species-poor mangrove ecosystems (compared with tropical and subtropical terrestrial forests) can enhance blue carbon storage. Our results identify (in line with our second hypothesis) niche complementarity[21] as one of the determinants of mangrove carbon pools in the Sundarbans Reserved Forest. Hence, stands with higher species richness exhibit higher carbon storage and vice versa. Mangroves hold their largest carbon storage in their

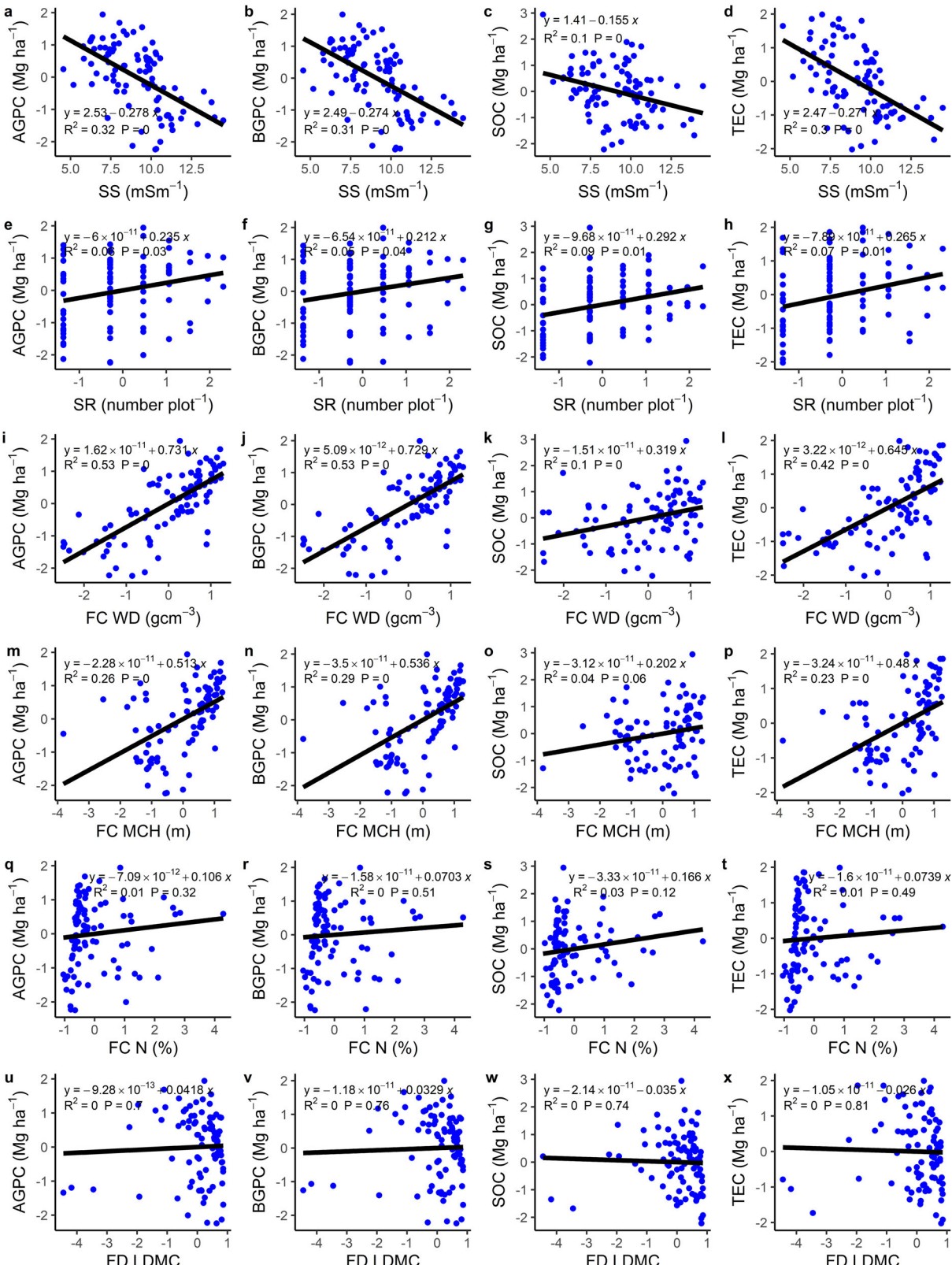

**Fig. 1 Bivariate relationships between sediment salinity (SS), species richness (SR), functional composition and diversity of different traits for hypothesized causal paths in the structural equation models. a–x** Sediment salinity (**a–d**), species richness (SR; **e–h**), functional composition wood density (FC WD; **j–l**), functional composition maximum canopy height (FC MCH; **m–p**), functional composition leaf litter nitrogen (FC N; **q–t**), functional diversity leaf litter dry matter content (FD; **u–x**) versus aboveground plant biomass carbon (AGPC), belowground plant biomass carbon (BGPC), sediment organic carbon (SOC) and total ecosystem carbon (TEC), respectively. Each blue point represents a sampling plot, while the black line indicates the fitted regression line. *P* value indicates the significance level.

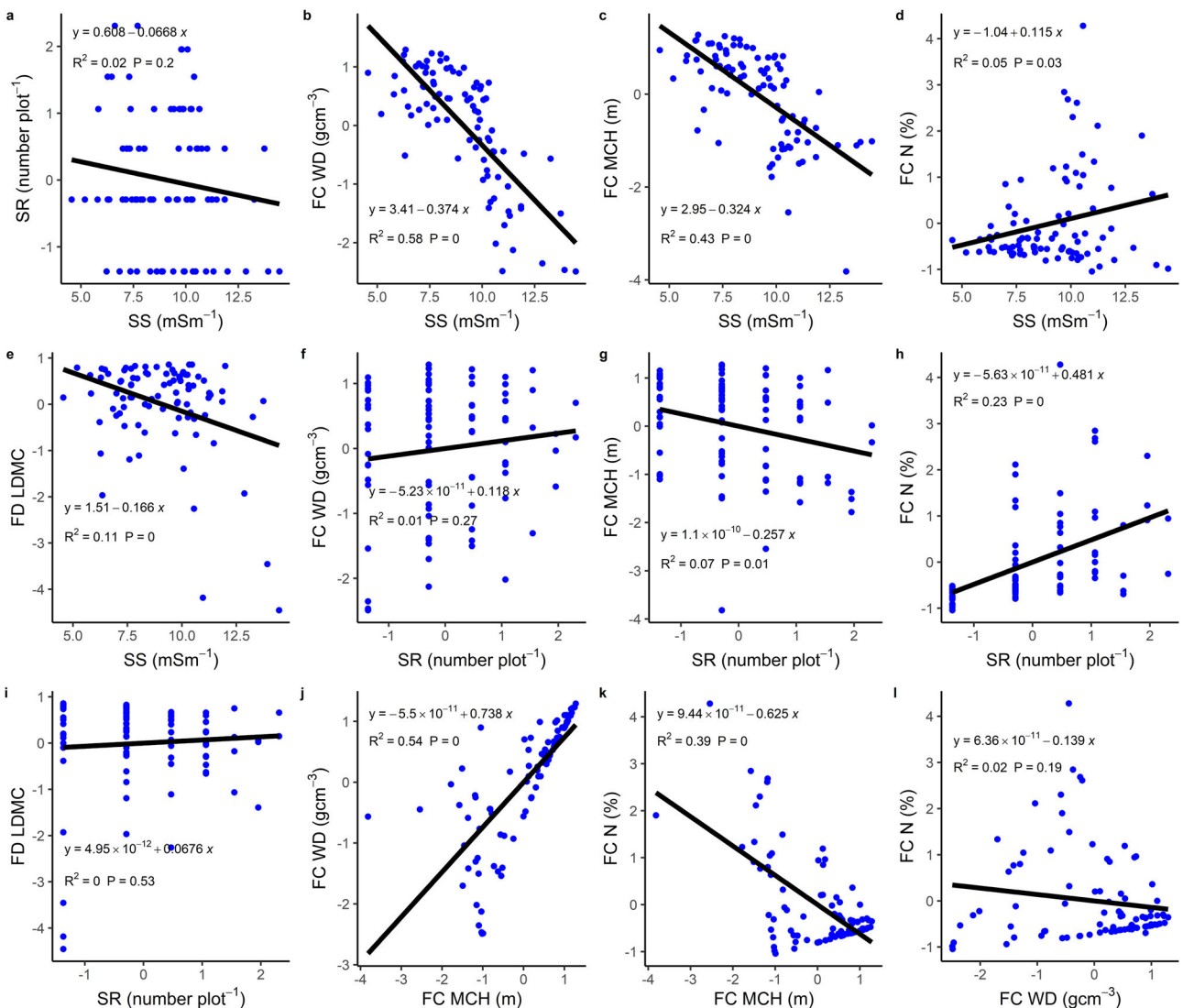

**Fig. 2 Bivariate relationships between sediment salinity (SS), species richness (SR), functional composition, and functional diversity of different traits for hypothesized causal paths in the structural equation models. a–l** Sediment salinity- versus species richness (**a**), -functional composition wood density (FC WD; **b**), -functional composition maximum canopy height (FC MCH; **c**), -functional composition litter nitrogen (FC N; **d**), -functional diversity leaf litter dry matter content (FD LDMC, **e**); species richness- versus functional composition wood density (**f**), -functional composition maximum canopy height (**g**), -functional composition litter nitrogen (**h**), -leaf litter dry matter content (**i**); functional composition maximum canopy height versus functional composition wood density (**j**); functional composition maximum canopy height versus functional composition litter nitrogen (**k**); functional composition wood density versus functional composition litter nitrogen (**l**). Each blue point represents a sampling plot, while the black line indicates the fitted regression line.

sediments[9] (11–98% of total ecosystem carbon stock depending on the site), especially in mixed stands[14]. In the present study, species diversity had a greater impact on sediment carbon than on other carbon pools, suggesting that maintaining a mixed species stand is beneficial when management objectives include increasing carbon storage in the sediment[13]. In addition, diverse stands also contribute to sediment retention through providing diverse trunk (buttress) and root systems (pneumatophores, prop and stilt roots, knee roots).

Second, the positive and stronger standardized association of functional composition than that of species diversity and functional diversity with different blue carbon storage reveal that the selection effect[21] (our third hypothesis) is the main driver of blue carbon storage. Thus, dominant species with high wood density and stature[22] and low nutrient retention capacity promote blue carbon storage in the Sundarbans Reserved Forest. Taking into account the negative relation between species richness and functional composition factor of maximum canopy height,

complementary resource-use strategies could be confined among dominant species[49].

Third, our findings may explain how indirect associations of species diversity mediate blue carbon storage in mangroves. A recent study in a South African mistbelt tropical forest[4] suggested that species richness exerted indirect associations with aboveground biomass carbon storage through both functional diversity and functional trait composition. However, in the present study, functional diversity proved less relevant. Hence, even relatively low functional diversity (e.g., of leaf litter dry matter content) can span the full range of traits (e.g., low and high wood density, maximum canopy height and litter nitrogen), rendering trait composition more relevant than trait diversity.

Fourth, the ecosystem carbon balance, through gains and losses of different carbon pools in above- and belowground plant biomass over time, determines whether the ecosystem is a sink[16] of atmospheric $CO_2$, and taller plant species play a crucial role in this regard. The strong predictability of multiple facets of

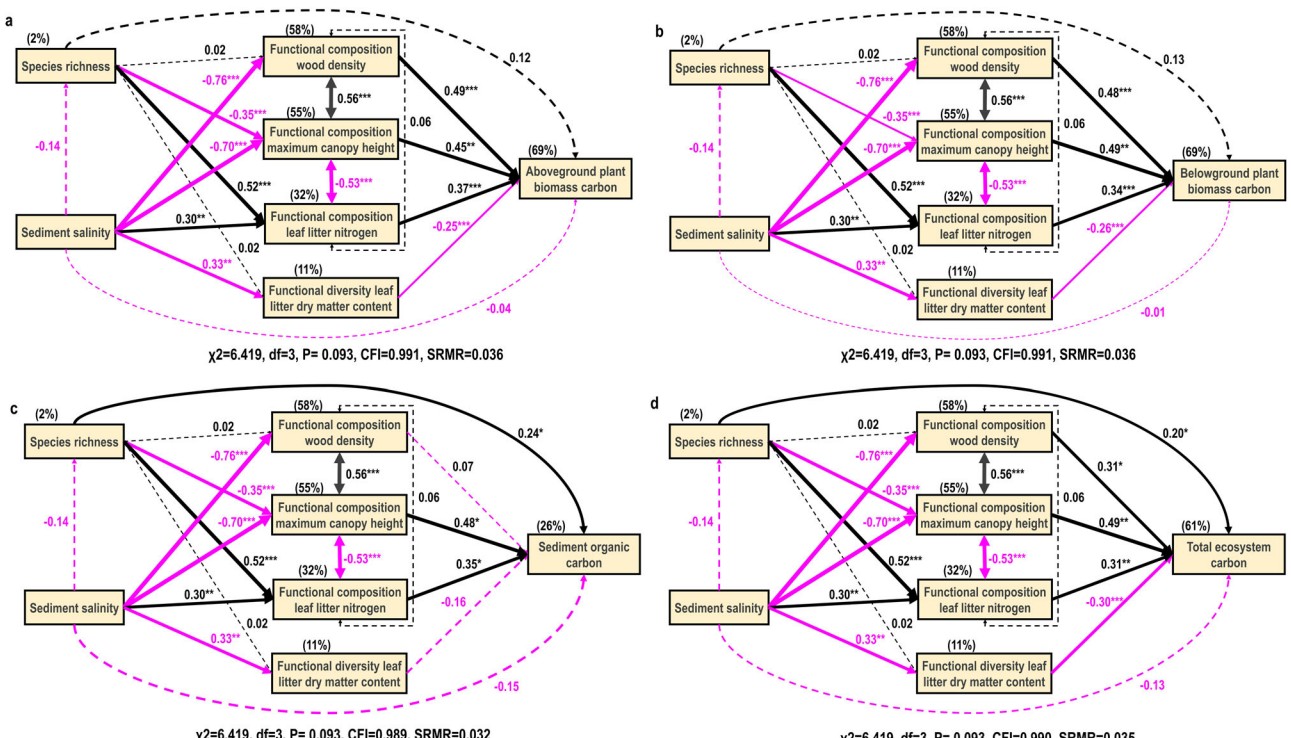

**Fig. 3 Structural equation models (SEMs) for understanding the diversity association with blue carbon storage. a–d** Aboveground plant biomass carbon (**a**), Belowground plant biomass carbon (**b**), Sediment organic carbon (**c**), and Total ecosystem carbon (**d**). All four SEMs had a similar nonsignificant $\chi^2$ (Chi-Square) of 6.83 ($P = 0.093$) with a comparative fit index close to one (CFI; 0.99) and standardized root mean square residual close to zero (0.036) indicating no significant deviation from model and datasets at 3 degree of freedom. The lines with pink and black indicate a negative and positive association between the two covariables. Arrows with numbers indicate the standardized association of predictors with dependent variables. Numbers with percentages above boxes independent variables indicate their explained variance (Coefficient of determinant: R squared) by all the predictors. The paths value with star mark indicate their significance level (***:$P < 0.001$, **:$P < 0.01$, *:$P < 0.05$) while the insignificant paths were indicated as dotted line ($P > 0.05$).

**Table 1 Indirect and total standardized association of sediment salinity and species richness on carbon pools.**

| Indirect and total association pathways | Aboveground plant Biomass carbon | | Belowground plant biomass carbon | | Sediment organic carbon | | Total ecosystem carbon | |
|---|---|---|---|---|---|---|---|---|
| | std.as | p value | std.as | p value | std.as | p value | std.as | p value |
| Indirect association of sediment salinity through species richness | −0.017 | 0.303 | −0.017 | 0.298 | −0.033 | 0.265 | −0.028 | 0.249 |
| Indirect association of sediment salinity through FD leaf litter dry matter content | 0.082 | 0.011 | 0.086 | 0.010 | 0.055 | 0.128 | 0.100 | 0.008 |
| Indirect association of sediment salinity through FC wood density | −0.371 | 0.000 | −0.366 | 0.000 | 0.053 | 0.721 | −0.234 | 0.032 |
| Indirect association of sediment salinity through FC maximum canopy height | −0.318 | 0.003 | −0.342 | 0.002 | −0.339 | 0.035 | −0.348 | 0.003 |
| Indirect association of sediment salinity through FC leaf litter nitrogen | 0.112 | 0.011 | 0.104 | 0.014 | 0.107 | 0.053 | 0.095 | 0.026 |
| Indirect association of sediment salinity through species richness and FD leaf litter dry matter content | −0.011 | 0.248 | −0.012 | 0.247 | −0.007 | 0.324 | −0.013 | 0.245 |
| Indirect association of sediment salinity through species richness and FC wood density | 0.050 | 0.222 | 0.050 | 0.223 | −0.007 | 0.731 | 0.032 | 0.267 |
| Indirect association of sediment salinity through species richness and FC maximum canopy height | 0.043 | 0.235 | 0.046 | 0.231 | 0.046 | 0.270 | 0.047 | 0.236 |
| Indirect association of sediment salinity through species richness and FC leaf litter nitrogen | −0.015 | 0.248 | −0.014 | 0.252 | −0.014 | 0.281 | −0.013 | 0.263 |
| Indirect association of species richness through FD leaf litter dry matter content | −0.006 | 0.823 | −0.006 | 0.823 | −0.004 | 0.824 | −0.007 | 0.823 |
| Indirect association of species richness through FC wood density | 0.007 | 0.826 | 0.007 | 0.826 | −0.001 | 0.852 | 0.005 | 0.827 |
| Indirect association of species richness through FC maximum canopy height | −0.159 | 0.008 | −0.171 | 0.006 | −0.170 | 0.048 | −0.174 | 0.009 |
| Indirect association of species richness through FC leaf litter nitrogen | 0.193 | 0.001 | 0.180 | 0.002 | 0.184 | 0.029 | 0.164 | 0.009 |
| Total association of sediment salinity | −0.444 | 0.000 | −0.466 | 0.000 | −0.141 | 0.216 | −0.362 | 0.000 |
| Total association of species richness | 0.159 | 0.068 | 0.137 | 0.119 | 0.254 | 0.009 | 0.191 | 0.028 |

The indirect and total associations of sediment salinity and species richness were based on structural equation models. The significant standardized associations (std.as) have p value < 0.05.
*FD* Functional diversity, *FC* Functional composition.

biodiversity of biomass carbon, ecosystem carbon, and even a substantial portion of sediment carbon in the present study suggests that plant diversity sustains carbon flows by influencing the functional composition of the community and, thereby, sediment carbon balance[18] and persistence for a longer time through litter input. Our results suggest that it is mostly a loss in functional distinctiveness (trait composition of wood density, maximum canopy height and leaf litter nitrogen) of mangroves that will have negative associations with blue carbon storage and, hence, on the potential of mangroves for mitigating climate change.

Fifth, in terms of policy implications, our findings demonstrate that ecosystems with high carbon storage-capacity and biodiversity can provide the co-benefits of both climate change mitigation and biodiversity conservation[50], and thus, meet both REDD + and Post-2020 biodiversity policies. Along this line, our findings indicate that the conservation of a mangrove forest of high species richness that covers the full range of traits (such as wood density, maximum canopy height and litter nitrogen) will be more efficient in capturing and storing carbon in plant biomass and sediment than replanting mono-specific mangrove stands. Beyond this, however, our findings also have further implications for rehabilitation or afforestation efforts. Even simple mangrove communities with low functional diversity will be efficient upon implementation, if the few implemented species differ sufficiently in their functional traits, i.e., they exhibit high functional distinctiveness. Following the concept of Ecosystem Design[51] for rehabilitation, selecting a combination of only few site-specific species with distinct traits, such as wood density, maximum canopy height and litter nitrogen, for instance, might yield the same results as trying to reestablish a highly diverse community, while being more feasible and more promising with respect to rehabilitation or afforestation success.

Finally, the indirect negative impact of sediment salinity on functional composition (the main driver of blue carbon storage) and, therefore, on blue carbon storage raises concerns about the adverse associations of rising sea levels[37,38] and the disruption of natural water flow[40]. Mangroves typically receive freshwater through upstream rivers, importing nutrients and lowering salinity levels and, thus, increasing the productivity of the stand and the storage of blue carbon[37,39]. On the other hand, rising sea levels will increase the inundation frequency in currently less-saline mangrove stands that are usually dominated by tall species with high wood density but low salt-tolerance[45]. Hence, sea level-rise will not only alter community composition[39,52] toward salt-tolerant species with lower wood density and canopy height[53], but also reduce the functional distinctiveness of wood density and maximum canopy height, thus diminishing the blue carbon storage capacity. In the current study we did not directly measure salinity at each vegetation plot. The plot-level salinity was obtained by a spatial interpolation of salinity measurements at 32 locations, which were randomly distributed in the same area as our vegetation plots. Although the spatial interpolation gave a good accuracy the small bias due to the interpolation might slightly affect the results of structural equation models, such as the weak relationship between salinity and blue carbon pools.

## Methods

**Field inventory data source and carbon pools assessment**. We used Sundarbans Reserved Forest inventory data, which were collected in 2009–2010 from 150 plots spanning over the whole SRF. Each plot was 4 min latitudinal and 2 longitudinal away from each other and composed of five 10 m radius nested circular subplots[45]. We used 90 (90 × 5 subplots) plots out of these 150 plots in our current study. The excluded 60 plots had at least one subplot that was occupied by a canal or river for more than 10% of its total area, or was completely underwater, or was severely disturbed by cyclone or fire. For diversity assessment, we used living tree data (stem diameter at breast height (DBH ≥ 10 cm) per plot, while for total ecosystem carbon we included live and dead carbon mass of trees and saplings, seedling carbon,

non-tree shrubs and herbs, and sediment organic carbon[45]. We used a species-specific allometric equation to calculate aboveground biomass of *Excocaria agallocha*[48] and a common allometric equation for all other species[46]. We also used a common allometric equation for belowground biomass of all trees[47]. Sapling (DBH ≤ 10) biomass was measured similar to trees. For non-timber vegetation and seedlings, destructive methods were followed for estimating biomass[45]. We converted the dry biomass of trees, understory, and down wood to carbon mass by multiplying by 0.5, as forest biomass contains half carbon by mass[25]. Sediment carbon was estimated at 1 m depth with two depth range: at the midpoint of 0–30 (15–20 cm) and 30–100 (60–65 cm) depth intervals from each of the five subplots. Bulk density was estimated after air-drying (in the field), and oven-dried to constant mass at 60 °C (to stop microbial decomposition) at the Khulna Integrated Protected Area Co-Management cluster office for determining bulk density. Sediment samples were further oven-dried at 105 °C Bangladesh Research Institute, Chittagong, Bangladesh. Walkley–Black's wet oxidizing method[54] was used for determination of organic carbon concentration[45]. We grouped carbon as above- and below -ground of live tree, sediment organic carbon and total ecosystem carbon storage (live and dead tree and sapling, seedling, non-tree shrubs and herbs and sediment organic carbon)[45].

**Functional traits and diversity metrics**. We used wood density, leaf photosynthesis rate, maximum canopy height, and four leaf-litter qualities (carbon; nitrogen; carbon to nitrogen carbon ratio; and leaf dry matter content) as functional traits. Maximum canopy height for each tree was extracted from 1997 forest inventory data in the Sundarbans Reserved Forest (Supplementary Table 1) while the wood density data was collected from the Bangladesh Forest Research Institute[55]. We used leaf trait values that were specific to Sundarbans[36,56]. The values of leaf litter traits[36] were averaged from data representing three different seasons (monsoon, pre-monsoon and post monsoon) (Supplementary Table 1). For species (relative abundance species < 2%) with missing traits, we used the average trait value (bold value; Supplementary Table 1). We assessed species richness by counting the number of species in each plot, while plot-wide diversity was calculated by Shannon diversity index (Eq. (1))[57]. Functional composition of traits was calculated using community weight mean (Eq. (2)), while for functional diversity computation we used the functional dispersion (Eq. (3)) metric in the FD package in R[58]. Both metrics are abundance weighted and the species wise total basal area in each plot was used as abundance because species contribution in an ecosystem is best represented by basal area[59]. As recommended, we standardized all the traits before calculating community weighted mean and functional dispersion.

$$H = \sum_{i=1}^{S}[P_i \times \ln(P_i)] \qquad (1)$$

where $H$ is the Shannon diversity for a plot. $S$ is the number of species (species richness), $P_i$ is the proportional of individuals of species $i$ in the plot.

$$CWM_{tj} = \sum_{i=1}^{S} P_{ij} t_i \qquad (2)$$

where $CWM_{tj}$ is the community weighted mean of trait $t$ for $j$ plot, $S$ is the species richness, $P_{ij}$ is the proportional of relative basal area of species $i$ for $j$ plot, and $t_i$ is the mean value of trait of species $i$.

$$FDis = \frac{\sum a_j Z_j}{\sum a_j} \qquad (3)$$

where $a_j$ is the abundance of species $j$ in terms of basal area and $Z_j$ is the distance of species $j$ to the weighted centroid $c$ which represents the centroid of the $n$ species in trait space.

**Sediment salinity and nutrients**. We followed the Evidence Density Estimation interpolation method[60] for generating spatial distribution maps of surface layer (0–15 cm) sediment salinity and sediment nutrient index (measured as a function of total organic matter, total nitrogen and total phosphorus) using data from 32 field sites[61]. These field sites spanned the whole Sundarbans Reserved Forest and were collected from 2010 to 2012. From these spatial maps, we extracted our subplot-wise (five subplots for each plot) sediment salinity and sediment nutrient index. Then we averaged the five subplot values of sediment salinity and sediment nutrient index for getting a plot -wide mean value.

**Statistical analysis**. We log transformed and standardized all four carbon pools, species richness, Shannon diversity index, functional composition and functional diversity of all traits to better meet linearity assumptions[44]. Functional diversity and functional composition of different traits were highly correlated (Supplementary Figs. 1 and 2). To remove the highly correlated variables from both functional diversity and functional composition of different traits, we applied multiple linear regressions for quantifying the variance inflation factors (a way to measure multicollinearity) using a threshold value of 10 (ref. [62]). We employed multiple linear regression (total ecosystem carbon storage was used as the response variable), starting with including all the covariates and incrementally removing the variables with highest variance inflation factor until achieving the threshold value (all covariates with a variance inflation factor below 10). This multicollinearity test resulted in functional diversity being composed of maximum canopy height, leaf

litter nitrogen and leaf litter dry matter content; and in functional composition being composed of wood density, maximum canopy height and leaf litter nitrogen (Supplementary Table 3 and Table 4). We then employed structural equation models using the Lavaan Package[63] in R for assessing the direct, indirect and total associations of sediment salinity, sediment nutrient index and multiple facets of biodiversity with different blue carbon pools[44,64].

We constructed structural equation models that are based on theories relating diversity and ecosystem function in the literature, that have been tested in terrestrial ecosystems[4]. We particularly focused on niche complementarity and selection theories where species richness and functional diversity of traits represented the former theory while functional composition of different traits represent the latter theory. We hypothesized that (i) Sediment salinity has direct and indirect associations with different blue carbon storage sediment nutrients, species richness and both functional composition and functional diversity, (ii) species diversity also has direct and indirect associations with blue carbon storage through functional composition and functional diversity of different traits[4], and (iii) functional composition has a greater association with different blue carbon storage than the two other biodiversity components[4]. Based on these hypothesizes, we tested a total of 32 structural equation models in different combinations of species diversity, functional composition and functional diversity of different traits (variance inflation factor value below 10; Supplementary Table 3 and 4) along with sediment nutrient and salinity (Supplementary Table 5). For every carbon pool, we first started with all the covariates, then gradually removed the hypothesized path with the highest $p$ value and continued until the model satisfied the model fitting measures. Of these 32 structural equation models, the combination of sediment salinity, species richness, functional composition of wood density, maximum canopy height and litter nitrogen, and functional diversity of leaf litter dry matter content exhibited the best fit (satisfied model fitting measures) with our dataset for aboveground and belowground plant biomass carbon, sediment organic carbon and total ecosystem carbon storage (Supplementary Table 5). We estimated the indirect association of sediment salinity on carbon pools by multiplying the standardized associations of all the paths from sediment salinity to species richness to functional composition and functional diversity to carbon pools[64]. For estimating the indirect associations of sediment salinity and species richness, we multiplied the standardized associations of all the paths from them to carbon pools through different mediators[64]. The total association of sediment salinity and species richness on carbon pools was calculated by adding the standardized direct and the standardized indirect associations[64].

We assessed the performance of structural equation model by using different fit indices that are widely applied for structural equation model. For example, an insignificant Chi-square test ($P > 0.05$; Eq. (4)), a comparative fit index (Eq. (5)) >0.90, and a standardized root mean square residual (Eq. (6)) close to 1 indicates no significant deviation of the structural equation model from the dataset[24,58,64]. We used the Lavaan package to calculate all the fit indices where they were in-built[58]. Finally, we also checked the multicollinearity of the final fitted models using a variance inflation factor test (Supplementary Table 6).

$$\chi^2 = (N-1)f \tag{4}$$

$$\text{CFI} = \frac{\max(\chi_0^2 - df_0) - \max(\chi_k^2 - df_k)}{\max(\chi_0^2 - df_0)} \tag{5}$$

$$\text{SRMR} = \sqrt{[\rho^{*-1}(e'W_s e)]} \tag{6}$$

Where, $\chi^2$: Chi-square test statistic; CFI: Comparative fit index; SRMR: Standardized root mean square residual; $N$: sample size; $f$: minimized discrepancy function; $0$: baseline model; df: degree of freedom; $k$: tested or hypothesized model, $\rho^*$: number of nonduplicated elements in the covariance matrix; $e$: a vector of residuals from a covariance matrix; $s$: a vector of the $\rho^*$ nonredundant elements in the observed covariance matrix; $W$: a weight matrix; $W_S$: a diagonal weight matrix used to standardize the elements of covariance matrix.

**Reporting summary**. Further information on research design is available in the Nature Research Reporting Summary linked to this article.

## Data availability
The datasets analyzed during the current study are available from the corresponding authors on reasonable request.

## Code availability
All the codes used to generate results, and to visualized the result during the current study are available from the corresponding authors on reasonable request.

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

## Acknowledgements
We are grateful to all the team members of the Sundarbans Reserved Forest Carbon Inventory who are not enlisted as coauthors in this study. Thanks to Steven Xu, for proofreading the paper. This research is supported by the National Key Research and Development Program of China (2017YFA0604300, 2018YFA0606500) and by the MEXT scholarship program of Ministry of Education, Culture, Sports, Science and Technology, Japan. United States Agency for International Development provides financial support to Bangladesh Forest Department for Sundarbans Reserved Forest Carbon Inventory 2009–2010. We also want to thank the Bangladesh Forest Department for sharing their data.

## Author contributions
R.M.M. conceived the study, R.M.M. and I.A. participated in field survey, R.M.M. and D.D. analyzed data, R.M.M., I.A., D.D., M.Z., M.K. and M.X. wrote the paper.

## Competing interests
The authors declare no competing interests.
