## [Peer Review File · Nature Communications]

REVIEWERS' COMMENTS

Reviewer #1 (Remarks to the Author):

I am satisfied that the authors have dealt adequately with my comments of the previous version. As it stands now, it is a superb manuscript that will have a major impact in the field of ecosystem conservation and carbon dynamics in relation to climate change mitigation.

Reviewer #2 (Remarks to the Author):

Thank you for meticulously addressing the comments in the revision. I note that the flow and consistency of the manuscript have been greatly improved.

Reviewer #4 (Remarks to the Author):

This revised version is much improved over the first version. The authors have addressed most of my major concerns, the figures and tables are more clear, and the paper overall has been significantly improved. The paper is one of the first papers that has examined traditional and more modern diversity indices (functional diversity and functional composition) to look at how different types of diversity influences carbon storage in mangrove forests. Their findings, that sediment salinity, species diversity, functional diversity and functional composition together explain 69%, 69%, 27% and 61% of the variation in above- and below-ground plant biomass carbon, sediment organic carbon and total ecosystem carbon storage, will be useful for resource managers, governments, and NGOs looking to conserve or restore mangrove forests. I also think their results will fuel future studies to try and look at how different types of diversity influence carbon stores as well as other ecosystem services, especially at a global level as the focus of this paper is on the Sundarbans. I would be curious to see if these patterns hold are or improved when additional data sets are included.

There are still two things that I would like to see addressed in the discussion. I didn't catch the first one the first time I reviewed this, but they used tree height from 1997, but biomass measurements from 2010. I assume that the same plots were measured in 1997 as in 2010 as (I think) these were part of Bangladesh's National Forest Inventory. The second is regarding the salinity data. Values from each plot were not measured, but were interpolated spatial distribution maps that were generated from 32 field sites. You need to add text in your discussion around salinity and tree height that mentions this. For example, the weak relationships you saw in your SEM between salinity and your C values could have been influenced by the fact that salinity was modeled OR maybe you would have seen stronger relationships among height and other factors if tree height had been collected in 2010 when biomass data was collected.

Lastly, is it the Sundarbans Reserve Forest or Reserved Forest?

Specific comments

L27-28 do you mean Sundarbans Reserve Forest instead of Reserved Forest?

L38 through increased carbon storage

L41 reducing atmospheric concentration of carbon dioxide.

L56 delete species richness because you later state functional diversity and functional composition
L56 both a direct and indirect
L58 add parentheses before i.e., and after community
L72 change to composition with carbon storage
L73 canopy height, specific leaf area,
L101 components of biodiversity: species richness, functional diversity and functional composition.

L102-103 place parentheses around i.e. statement

L103 consider changing to into modelling along with other traditional and modern diversity measures and sediment salinity...

L111 Reserve or Reserved?

L139 change to species diversity to species richness, that's what you report in Figure 1

L239 carbon in the present study? There is a word missing here

L242 can you list some examples of traits whose loss would have negative associations?

L275 delete the sentence For detailed inventory methods and just place the citation at the end of the first sentence of this paragraph (L272-273)

L276 what does water bodies > 10% mean? More than 10% of the plot was submerged?

Figure 1 and 2- double check all x and y axes so that they are the same (e.g., some say AGC, some say AGPC).

Figure 3 - what do pink versus black lines indicate?

L634 - arrows with numbers indicate

L634 - Numbers with percentage above boxes

L634 - what is the explained variance relative to? So for example, AGPC explained variance if 69%, is that relative to all of the factors that contribute to it? I am not sure what this means.

Rich MacKenzie

REVIEWERS' COMMENTS

Reviewer #1 (Remarks to the Author):

Comment: I am satisfied that the authors have dealt adequately with my comments of the previous version. As it stands now, it is a superb manuscript that will have a major impact in the field of ecosystem conservation and carbon dynamics in relation to climate change mitigation.

Response: Thank you for your positive remarks on our revised version. The foundation of our current improved version is solely your valuable comments. Thank you so much for your comments and suggestions.

Reviewer #2 (Remarks to the Author):

Comment: Thank you for meticulously addressing the comments in the revision. I note that the flow and consistency of the manuscript have been greatly improved.

Response: Thank you for the positive feedback on our revised manuscript. We are delighted that you are happy with the flow and consistency of our current version of manuscript.

Reviewer #4 (Remarks to the Author):

Comment: This revised version is much improved over the first version. The authors have addressed most of my major concerns, the figures and tables are more clear, and the paper overall has been significantly improved. The paper is one of the first papers that has examined traditional and more modern diversity indices (functional diversity and functional composition) to look at how different types of diversity influence carbon storage in mangrove forests. Their findings, that sediment salinity, species diversity, functional diversity and functional composition together explain 69%, 69%, 27% and 61% of the variation in above- and below-ground plant biomass carbon, sediment organic carbon and total ecosystem carbon storage, will be useful for resource managers, governments, and NGOs looking to conserve or restore mangrove forests. I also think their results will fuel future studies to try and look at how different types of diversity influence carbon stores as well as other ecosystem services, especially at a global level as the focus of this paper is on the Sundarbans. I would be curious to see if these patterns hold or are improved when additional data sets are included.

Response: Thank you for your positive remarks on our revised manuscript which improved a lot with your constructive and thoughtful comments and suggestion. I would definitely want to further look the biodiversity-carbon/multifunctionality relationship in mangrove at global scale, if I get access to global dataset. For example, Donato et al. 2011, Kauffman et al. 2020 and other recent dataset including your research groups dataset.

Comment: There are still two things that I would like to see addressed in the discussion. I didn't catch the first one the first time I reviewed this, but they used tree height from 1997, but biomass measurements from 2010. I assume that the same plots were measured in 1997 as in 2010 as (I think) these were part of Bangladesh's National Forest Inventory. The second is regarding the salinity data. Values from each plot were not measured, but were interpolated spatial distribution maps that were generated from 32 field sites. You need to add text in your discussion around salinity and tree height that mentions this. For example, the weak relationships you saw in your SEM between salinity and your C values could have been influenced by the fact that salinity was modeled OR maybe you would have seen stronger relationships among height and other factors if tree height had been collected in 2010 when biomass data was collected.

Response: Maximum tree height has been commonly used as a site index in forestry to describe the potential for trees to grow at a particular location or "site". In a natural forest the maximum tree height, therefore, barely changes over time if the site condition (e.g. climate and soil quality) does not change too much. In the current study the maximum tree height was derived based on the measurements of all the individual trees in 1202 plots surveyed in 1997. We did not remeasure the maximum tree height in 2010 because the maximum tree height is stable for a given species. Our assumption is supported by many studies in forestry where the site index, namely maximum tree height or average of dominant tree height in a stand, was assumed as a constant and also used as one of the key indicators of plant biomass and carbon (Stahl et al. 2014; Mensha et al. 2016; Gibert et al. 2016; Kunstler et al. 2016, Moreno-Martínez et al. 2018, Noordermeer et al. 2020; Wright et al. 2010).

Use TRY trait database (<https://www.try-db.org/TryWeb/Home.php>):

Kunstler, G., Falster, D., Coomes, D.A. *et al.* 2016 Plant functional traits have globally consistent effects on competition. *Nature*, **529**, 204–207.

Moreno-Martínez, Á.; Camps-Valls, G.; Kattge, J.; Robinson, N.; Reichstein, M.; van Bodegom, P.; Kramer, K.; Cornelissen, J.H.C.; Reich, P.; Bahn, M.; *et al.* **2018**. A methodology to derive global maps of leaf traits using remote sensing and climate data. *Remote Sens. Environ.* **218**, 69–88

Mensah, S., Veldtman, R., Assogbadjo, A. E., Glèlè Kakaï, R. & Seifert, T. 2016. Tree species diversity promotes aboveground carbon storage through functional diversity and functional dominance. *Ecol. Evol.* **6**, 7546–7557

Use published trait data:

Stahl et al. 2014. Distribution limits of North American trees. *Proceedings of the National Academy of Sciences*; **111** (38) 13739-13744; DOI: 10.1073/pnas.1300673111

Gibert et al. 2016. On the link between functional traits and growth rate: meta-analysis shows effects change with plant size, as predicted. *Journal of Ecology*, **104**, 1488–1503

Noordermeer et al. 2020. Predicting and mapping site index in operational forest inventories using bitemporal airborne laser scanner data. *Forest Ecology and Management*;457, 117768

Wright, S.J., Kitajima, K., Kraft, N.J.B., Reich, P.B., Wright, I.J., Bunker, D.E. et al. (2010) Functional traits and the growth–mortality trade-off in tropical trees. *Ecology*, 91, 3664–3674.

We agree with the reviewer that the lack of direct measurement of salinity at each vegetation plot may introduce bias to the SEM analysis. We used the salinity data measured at 32 locations and interpolated to the whole study area. The salinity at each vegetation plot was extracted from the interpolated salinity map. We don't think the interpolated data will affect our results considerably. First, we checked the accuracy of the interpolation by dropping 3 locations randomly from the original salinity dataset and fitting a spatial interpolation model with the rest 29 points. We found that the interpolated salinity matched the observed data very well (5.70, 9.20, and 11.71 vs. 5.68, 10.26, and 12.17, respectively; $R^2=0.98$). Second, the 32 salinity measurements were randomly distributed in the same area as our vegetation plots. The salinity at each vegetation plot might be slightly biased due to the interpolation. But the relationship/association with other variables, such as carbon pools, should be barely affected. Nevertheless, we added a few sentences (Lines 280-284) to discuss this issue as the Reviewer has suggested.

Comment: Lastly, is it the Sundarbans Reserve Forest or Reserved Forest?

Response: The official name is Sundarbans Reserved Forest (Bangladesh Sundarbans). However, in some publications authors wrote Reserve (Wrong) in place of Reserved.

Specific comments

Comment: L27-28 do you mean Sundarbans Reserve Forest instead of Reserved Forest?

Response: Please see previous response

Comment: L38 through increased carbon storage

Response: Thank you for this editing. We have added “increased” after through and before carbon.
Line:34

Comment: L41 reducing atmospheric concentration of carbon

dioxide. Response: “of carbon dioxide” has been added. Line 47

Comment: L56 delete species richness because you later state functional diversity and functional composition

Response: Agreed and deleted species richness.

Comment: L56 both a direct and indirect

Response: Article “a” has been added after both and before direct. Line: 63

Comment: L58 add parentheses before i.e., and after community

Response: Thank you. We have added parenthesis. Line 64-65

Comment: L72 change to composition with carbon storage

Response: We have replaced “to” with “with”. Line 79

Comment: L73 canopy height, specific leaf area,

Response: We have added a comma after canopy height. Line 80

Comment: L101 components of biodiversity: species richness, functional diversity and functional composition.

Response: “parenthesis” and “biotic factors” have been replaced with colon. Line 108

Comment: L102-103 place parentheses around i.e. statement

Response: Parenthesis has been added. Line 109-110

Comment: L103 consider changing to into modelling along with other traditional and modern diversity measures and sediment salinity...

Response: Thank you for this comment. We agreed and replaced species diversity with “other traditional and modern diversity measures” Line 110-111

Comment: Comment: L111 Reserve or Reserved?

Response: The official name of Bangladesh Sundarbans is Sundarbans Reserved Forest.

L139 change to species diversity to species richness, that’s what you report in Figure 1

Response: Thank you for this comment. We have changed species diversity to species richness. Line 149

Comment: L239 carbon in the present study? There is a word missing here

Response: Yeah, a preposition is needed here and “in” is perfectly ok here. We have added in before the. Line 249

Comment: L242 can you list some examples of traits whose loss would have negative associations?

Response: we have added the functional traits. Line 252-253

Comment: L275 delete the sentence For detailed inventory methods and just place the citation at the end of the first sentence of this paragraph (L272-273)

Response: This sentence has been deleted.

Comment: L276 what does water bodies > 10% mean? More than 10% of the plot was submerged?

Response: We have revised this sentence as “The excluded 60 plots had at least one subplot that was occupied by a canal or river for more than 10 % of its total area, or was completely underwater, or was severely disturbed by cyclone or fire.” Line 292-294

Comment: Figure 1 and 2– double check all x and y axes so that they are the same (e.g., some say AGC, some say AGPC).

Response: Thank for noting this. We have changed AGC to AGPC.

Comment: Figure 3 – what do pink versus black lines indicate?

Response: The lines with pink and black indicate negative and positive association between two covariables. We have explained it in figure caption.

Comment: L634 – arrows with numbers indicate

Response: We have corrected as you suggested.

Comment: L634 – Numbers with percentage above boxes

Response: “above” have been added before boxes.

Comment: Comment: L634 – what is the explained variance relative to? So for example, AGPC explained variance if 69%, is that relative to all of the factors that contribute to it? I am not sure what this means.

Response: Yeah, the explained variance is a contribution of all the

predictors. **Rich MacKenzie**